# Fine-tuned In-Context Learning Transformers are Excellent Tabular Data Classifiers

## Abstract

The recently introduced TabPFN pretrains an In-Context Learning (ICL) transformer on synthetic data to perform tabular data classification. In this work, we extend TabPFN to the fine-tuning setting, resulting in a significant performance boost. We also discover that fine-tuning enables ICL-transformers to create complex decision boundaries, a property regular neural networks do not have. Based on this observation, we propose to pretrain ICL-transformers on a new forest dataset generator which creates datasets that are unrealistic, but have complex decision boundaries. TabForest, the ICL-transformer pretrained on this dataset generator, shows better fine-tuning performance when pretrained on more complex datasets. Additionally, TabForest outperforms TabPFN on some real-world datasets when fine-tuning, despiting having lower zero-shot performance due to the unrealistic nature of the pretraining datasets. By combining both dataset generators, we create TabForestPFN, an ICL-transformer that achieves excellent fine-tuning performance and good zero-shot performance.

## 1 Introduction

Tabular data classification is widespread across all industries, leading to an increased interest in the research field of deep learning for tabular data (Liakos et al., 2018; Zhang et al., 2020; Keith et al., 2021; Pang et al., 2022). This type of classification involves classifying a target variable based on a set of attributes, which are commonly stored in tabular format. Examples of tabular classification include predicting the existence of chronic kidney disease based on blood test results (Ogunleye & Wang, 2020), estimating the click-through rate of advertisements (Richardson et al., 2007), and predicting the stability of pillars in hard rock mines (Liang et al., 2020). Despite the significance of tabular data, major breakthroughs in AI, as demonstrated in vision and language domains, have yet to reach the tabular domain. In fact, neural networks are currently outperformed by tree-based machine learning algorithms such as XGBoost (Chen & Guestrin, 2016) and CatBoost (Prokhorenkova et al., 2018) in tabular classification tasks (Gorishniy et al., 2021; Grinsztajn et al., 2022; McElfresh et al., 2023).

In an attempt to bridge this performance gap, a recent method called *tabular prior-data fitted networks* (TabPFN) (Hollmann et al., 2023) introduces an *in-context learning* (ICL) (Dong et al., 2023) scheme, demonstrating promising results (Grinsztajn et al., 2022). This tabular ICL-transformer can predict test observations zero-shot: with only one forward pass using training observations included in the context. Hollmann et al. generate their pretraining data synthetically, focusing on creating realistic datasets that act as a "prior". They make their datasets realistic by carefully crafting correlations between features, introducing variety in feature importance, and leveraging structural causal models to simulate causal relationships.

In this work, we extend TabPFN to the fine-tuning setting. We show that fine-tuning this ICL-transformer on downstream tasks boosts the performance, particularly on datasets with more than a thousand samples. We also find that increasing the context size provides a performance increase for both zero-shot and fine-tuning. Overall, we find that fine-tuning provides such a large performance boost that we recommend always using it over zero-shot when the number of observations exceeds a thousand, and using zero-shot only when inference speed is an issue.

During this process, we also discovered an interesting property of ICL-transformers. Fine-tuned ICL-transformers can create complex decision boundaries, see Figure 1 for an example. Intuitively,

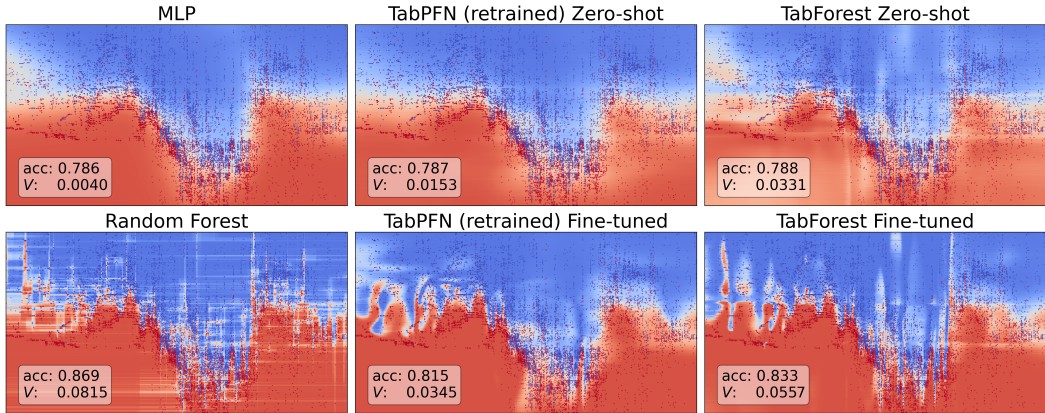

Figure 1: Comparison of decision boundaries for the Electricity dataset (OpenML ID 44156). Axis represent features, colors are predicted class probabilities, and dots are test observations. Fine-tuned variants show a higher complexity score $V$ (see section 5.3) than zero-shot variants.

the complexity is determined by how far the decision boundary differs from a simple linear line. Neural networks trained from scratch on tabular data often have overly simple decision boundaries, a phenomenon known as simplicity bias (Shah et al., 2020), while tree-based methods do not suffer from this (Grinsztajn et al., 2022). We find that fine-tuned ICL-transformers, in contrast, are able to create these complex decision boundaries similar to tree-based methods.

Given this observation, we wonder whether pretraining on more complex data would improve the fine-tuning performance. To this end, we introduce the novel forest dataset generator, which creates highly complex synthetic datasets using a simple decision tree. By varying the parameters of the decision tree, we can control the complexity of the generated datasets. We observe that TabForest, the ICL-transformer pretrained on this forest dataset generator, achieves better fine-tuning performance as the complexity of the generated datasets increases.

Furthermore, TabForest shows fine-tuning performance that surpasses TabPFN on specific real-world datasets, even though the zero-shot performance of TabForest is significantly lacking compared to TabPFN. This suggests that for the fine-tuning performance of some real-world datasets, pretraining the ICL-transformer on highly complex datasets is more important than pretraining on realistic datasets, although for zero-shot, we would always prefer the TabPFN dataset generator.

As we would like to have a single ICL-transformer that performs well across all real-world tabular datasets, we mix the TabPFN and forest dataset generators to pretrain TabForestPFN. This model has excellent fine-tuning performance on two benchmarks (Grinsztajn et al., 2022; McElfresh et al., 2023), matching the performance of either TabForest or TabPFN. At the same time, mixing in the forest dataset generator does not seem to harm the zero-shot performance at all. This makes TabForestPFN the preferred ICL-transformer over TabForest and TabPFN.

In conclusion, fine-tuned ICL-transformers are highly effective tabular data classifiers, capable of creating complex decision boundaries. This new insight advances our understanding of tabular ICL-transformers and opens up new avenues for further research to enhance their performance. With further developments, we anticipate a significant shift in the field of tabular data, moving from tree-based methods towards ICL-transformers.

## 2 RELATED WORKS

There are three main branches of tools for tabular data classification: classical statistical methods like linear regression, K-nearest neighbors, Gaussian processes (Williams & Rasmussen, 1995), and support vector machines (Hearst et al., 1998); tree-based algorithms like XGBoost (Chen & Guestrin, 2016), CatBoost (Prokhorenkova et al., 2018), and LightGBM (Ke et al., 2017); and neural network-based methods such as the approach presented in this paper. There are several papers benchmarking the different methods (Gorishniy et al., 2021; Shwartz-Ziv & Armon, 2022; Grinsz-

tajn et al., 2022; McElfresh et al., 2023; Zabërgja et al., 2024). Overall, tree-based methods stand at the top, with neural networks ranging from inferior to at best competitive.

Nonetheless, there have been numerous approaches that tackle tabular data classification with neural networks. First, we have the class of neural networks trained *from scratch*: training starts from random initialized weights and is only trained on the data at hand. Research has focused on architectures (Katzir et al., 2021; Somepalli et al., 2021; Arik & Pfister, 2021; Gorishniy et al., 2023; Huang et al., 2020; Chen et al., 2023a), embeddings (Ruiz et al., 2023; Gorishniy et al., 2022; Chen et al., 2023b), and regularization (Shavitt & Segal, 2018; Kadra et al., 2021).

In general, methods training from scratch can struggle because tabular datasets can be small. So, researchers have sought ways to use large volumes of tabular data or to change the training objective. Some employ self-supervised learning (Kossen et al., 2021; Yoon et al., 2020; Zhu et al., 2023; Bahri et al., 2022; Ucar et al., 2021; Sui et al., 2023), or closely related transfer learning techniques (Nam et al., 2023; Levin et al., 2023; Zhou et al., 2023). Others leverage pretrained LLMs or language data (Hegselmann et al., 2023; Zhang et al., 2023; Kim et al., 2024; Yan et al., 2024) to make predictions.

One of those related transfer learning methodologies is *tabular in-context learning*, a new field sparked by TabPFN (Hollmann et al., 2023). Currently, there is ongoing research on how to scale TabPFN to encompass more observations and features (Ma et al., 2023; Feuer et al., 2024; Thomas et al., 2024), as this architecture is limited by GPU memory. Our fine-tuning work can be seen as one approach to tackle this issue.

## 3 Preliminaries

In tabular classification, we are interested in predicting targets $y \in \mathbb{N}$ given features $\boldsymbol{x} \in \mathbb{R}^d$, where $d$ is the number of features. We predict $y$ using an *in-context learning* (ICL) transformer pretrained on a synthetic dataset. The in-context learning allows the transformer to predict targets based on other observations included in the forward pass. In our work, *zero-shot* refers to one forward pass through the ICL-transformer without any fine-tuning, while *fine-tuning* refers to one forward pass through the ICL-transformer after fine-tuning. Our work builds on TabPFN (Hollmann et al., 2023), so below we explain their dataset generator and their transformer architecture.

### 3.1 TabPFN Dataset Generator

The TabPFN authors create their own synthetic dataset using *Bayesian Neural Networks* (BNN) and *Structural Causal Models* (SCM). To construct a dataset, they first create a BNN or a SCM with random characteristics and with randomly initialized weights. Then they randomly draw an input $\boldsymbol{X}$ and pass it through the model to generate output $\boldsymbol{y}$. Their final dataset is given by $(\boldsymbol{X}, \boldsymbol{y})$. See their paper (Hollmann et al., 2023) for more details.

In their approach, they emphasize their ability to create realistic datasets, and even call their generator a "prior". They chose SCMs specifically because it can capture real-world causal mechanisms. One other aspect they focus on is simplicity, biasing the generator towards less complex input-output relationships. Additionally, they ensure the inputs $\boldsymbol{X}$ have natural correlations by correlating the features blockwise, and they vary their feature importance by tuning the magnitude of weights belonging to different features. These methods suggest the authors believe creating realistic datasets is important for achieving good performance.

### 3.2 Architecture

In our work, we use the architecture from TabPFN, and make no changes to isolate the effect of the dataset generator. This ICL-transformer has as input the features $\boldsymbol{X}_{support} \in \mathbb{R}^{|S| \times d_f}$ and targets $\boldsymbol{y}_{support} \in \mathbb{N}^{|S|}$ from *support* set $S$ and features $\boldsymbol{X}_{query} \in \mathbb{R}^{|Q| \times d_f}$ from *query* set $Q$. The output is a prediction for $\boldsymbol{y}_{query} \in \mathbb{R}^{|Q|}$. The query set $Q$ represents the observations we want to predict, while the support set $S$ includes the observations we base our prediction on. This architecture accepts a fixed number of features $d_f$, see also the preprocessing discussed in Appendix A.3.

In this transformer, a token with dimension $d_{token}$ represents all features of a single observation. The creation of support tokens $H_{support} \in \mathbb{R}^{|S| \times d_{token}}$ and query tokens $H_{query} \in \mathbb{R}^{|Q| \times d_{token}}$ is

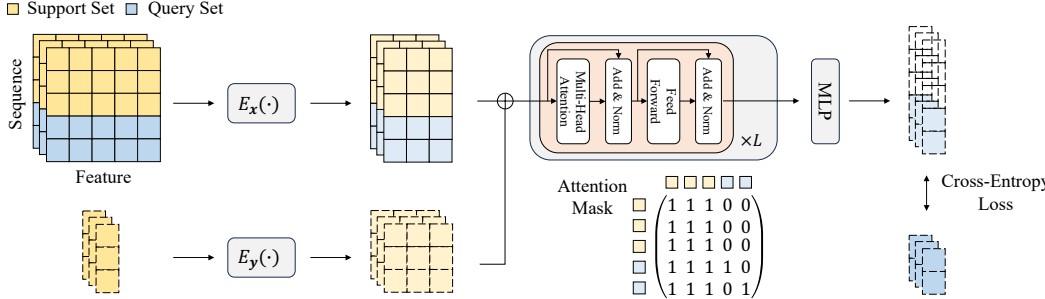

Figure 2: Base ICL-transformer architecture. On the left, dataset features and targets are separately encoded into tokens. On the right, the targets of the query dataset are used as label. In the middle is the ICL-transformer with the attention mask.

given by equations 1 and 2.

$$\boldsymbol{H}_{support} = \boldsymbol{X}_{support}\boldsymbol{W}_x + \boldsymbol{y}_{support}\boldsymbol{w}_y^T \tag{1}$$

$$\boldsymbol{H}_{query} = \boldsymbol{X}_{query}\boldsymbol{W}_x \tag{2}$$

Here, we embed the features linearly using weights $\boldsymbol{W}_x \in \mathbb{R}^{d_f \times d_{token}}$. Input classes $\boldsymbol{y}_{support}$ are also embedded using a linear layer with weights $\boldsymbol{w}_y \in \mathbb{R}^{d_{token}}$, in which $\boldsymbol{y}_{support}$ is treated as a float. Biases are used but omitted in the equations for conciseness. In Figure 2, $E_x$ refers to the multiplication with $\boldsymbol{W}_x$ and $E_y$ represents the product with $\boldsymbol{w}_y$.

After the embedding, we push the tokens through a standard transformer architecture with a special attention mask. Support tokens are only able to see other support tokens, and query tokens can only see all support tokens and themselves, with no attention to other query tokens. This attention mask ensures the prediction of an observation does not depend on which other test observations are included in the forward pass. The complete architecture is given in Figure 2.

## 4 METHODOLOGY

The full tabular data classification pipeline is given by: synthetic data generation (3.1 and 4.1), data preprocessing (A.3), architectural design (3.2) and fine-tuning (4.2). In this section, we introduce our new dataset generator and our proposed fine-tuning procedure.

### 4.1 FOREST DATASET GENERATION

Our goal is to create a simple dataset generator that produces datasets with complex patterns to train on, in contrast to the TabPFN (Hollmann et al., 2023) generator that aims to create realistic datasets. This forest dataset generator will better enable ICL-transformers to create complex decision boundaries. We base our dataset generator on decision trees, because of their ability to create highly complex decision boundaries with minimal computational cost. The idea is to *overfit* the decision tree to randomly generated features and targets. This fitted decision tree is then used as a data-generating process. See Algorithm 1 for the method and Figure 3 for examples of generated data.

Table 1: Hyperparameters for the Forest Dataset Generator

| Hyperparameter | min | max |
|---|---|---|
| base size | 1024 | 1024 |
| dataset size | 128 | 1024 |
| tree depth | 1 | 25 |
| number of features | 3 | 100 |
| number of classes | 2 | 10 |
| ratio of categorical features | 0.0 | 1.0 |

Our forest dataset generator allows datasets to vary in the number of classes, observations, numerical features, and categorical features. There are two parameters that contribute to the decision boundary complexity. The *base size* is the number of observations used to fit the decision tree; more observations means more places for the decision tree to split on. The *tree depth* determines how deep the decision tree will go before exiting the fitting algorithm, with higher depth leading to increased complexity.

---

**Algorithm 1** Forest Dataset Generation

---

**Require:** *n_classes, n_features, base_size, dataset_size, tree_depth, categorical_perc*
    **Draw** $X \sim \mathcal{N}(base\_size, n\_features)$
    **Draw** $y \sim \mathcal{N}(base\_size)$
    **Fit** a decision tree on $(X, y)$ of depth *tree_depth*.
    **Draw** $X_2 \sim \mathcal{N}(dataset\_size, n\_features)$
    **Convert** *categorical_perc* features of $X_2$ to categorical.
    **Predict** $y_2$ using the decision tree on $X_2$.
    **Transform** $y_2$ using quantile transformation to uniform.
    **Discretize** $y_2$ into *n_classes* classes.
**Ensure:** $(X_2, y_2)$

---

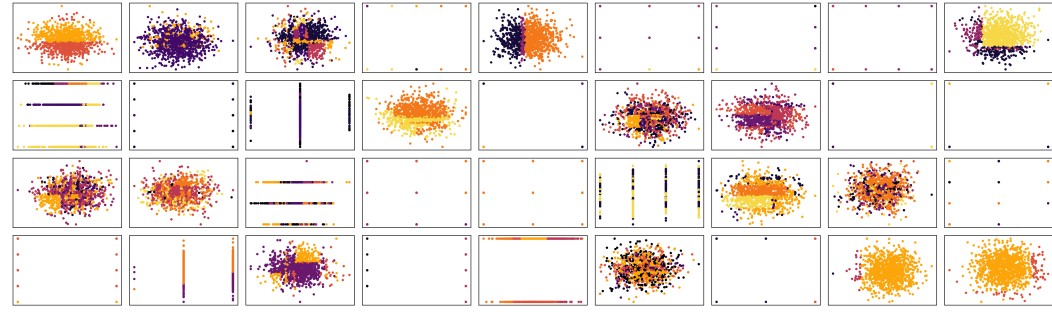

Figure 3: Generated forest data. Every box is a generated dataset with its own classes (color) and features (axes). The data clouds look unrealistic: decision boundaries are always orthogonal, and there is no feature correlation. Generated with base size of 1024, dataset size of 1024, maximum tree depth between 1 and 25, two features, and between 2 and 10 number of classes.

For every new synthetically generated dataset, we uniformly draw the hyperparameters from the bounds shown in Table 1. Because both hyperparameters influence the complexity, we decided to keep the base size fixed. In the final step of the algorithm, the targets $y_2$ are discretized by uniformly drawing bucket boundaries between 0.0 and 1.0, and assigning a class to each bucket, creating varying degrees of class imbalance.

## 4.2 FINE-TUNING PROCEDURE

In our work, we introduce fine-tuning to the tabular ICL-transformer. When fine-tuning, we like to draw support and query sets from our training data such that the performance generalizes to the test set. This requires careful consideration of dataset splitting. The benchmark datasets already provide us with a training-validation-test split. We use this validation dataset for early stopping.

Every gradient descent step, we randomly draw a 80/20 support and query split from the training dataset. For validation, we draw the support set from the training set and draw the query set from the validation set. Every validation sample is seen exactly once, while the support samples are randomly drawn with replacement. We use early stopping based on the validation loss, which is calculated after every fine-tuning step.

The early stopping technique can decide to stop fine-tuning immediately if the validation loss increases in the first step of training, which allows us to fall back on the zero-shot performance in case fine-tuning harms the performance. This is especially important when using very small datasets, as they are prone to overfitting. At the same time, fine-tuning can leverage all samples in training dataset, while zero-shot cannot use more samples than that fit on the GPU.

## 5 EXPERIMENTS

In our experiments we consider five pre-trained models, each with a zero-shot and a fine-tuned version:

- **TabPFN (original)** is the original implementation by the TabPFN (Hollmann et al., 2023) authors, fine-tuned by us. The weights are downloaded from their GitHub.
- **TabPFN (retrained)** is our implementation of TabPFN, trained by us on the TabPFN-dataset.
- **TabForest** is trained by us on our forest dataset.
- **TabForestPFN** is trained by us on both the TabPFN-dataset and our forest dataset.
- **TabScratch** is not pretrained. We include this as a baseline.

Comparing the behavior and performance allows us to understand the effect of the different synthetic datasets. Training and hyperparameter settings are given in appendix A.4, benchmarks used and results obtained are given below.

## 5.1 INTRODUCTION OF THE BENCHMARK DATASETS

We show the results of our pre-trained architectures on two benchmarks, tested against publicly available results provided by the authors of the benchmarks. We include all their tested methods and datasets where possible. Appendix A.5 lists all used datasets.

The TabZilla (McElfresh et al., 2023) benchmark tests 20 algorithms on 176 classification datasets with sizes ranging from 32 observations to over a million. We selected 94 out of 176 datasets, see appendix A.5. The medium-sized benchmark which we refer to as WhyTrees (Grinsztajn et al., 2022) consists of 23 classification datasets with 2923 to a maximum of 10,000 observations. The benchmark is split into 7 datasets with only numerical features and 16 datasets with both numerical and categorical features.

Both benchmarks perform random hyperparameter search on their algorithms. TabZilla runs up to 30 times per algorithm and WhyTrees runs a few hundred times, up to 2500 runs. The ICL-transformers run only on default settings because we noticed little gains in performance when changing the fine-tuning hyperparameters.

## 5.2 MAIN RESULTS OF TABFORESTPFN

The results on the TabZilla benchmark are shown in Table 3, see Appendix A.7 for alternative presentations. For the WhyTrees benchmark, the comparison of fine-tuned TabForestPFN with the benchmark algorithms is shown in Figure 4, and the comparison with other ICL-transformer variants in Table 2, while results on individual datasets can be seen in Appendix A.8. We present the running time of TabForestPFN on all datasets in Appendix A.6.

In these figures and tables, we see that fine-tuning considerably improves the performance of ICL-transformers. Where the zero-shot variants perform very mediocre compared to XGBoost and the other baselines, the fine-tuned variants are extremely competitive. Given the poor performance of TabScratch, we can clearly see that both the pretraining and the fine-tuning are important.

Table 2: WhyTrees Results. Normalized accuracy for mixed and numerical features as shown in Figure 4.

|  | Mixed | Numerical |
|---|---|---|
| Zero-shot | | |
| TabScratch | 0.000 | 0.000 |
| TabPFN (original) | **0.534** | 0.624 |
| TabPFN (retrained) | 0.481 | 0.635 |
| TabForest | 0.388 | 0.536 |
| TabForestPFN | 0.473 | **0.655** |
| Fine-tuned | | |
| TabScratch | 0.481 | 0.570 |
| TabPFN (original) | 0.742 | 0.775 |
| TabPFN (retrained) | 0.775 | 0.817 |
| TabForest | **0.853** | **0.854** |
| TabForestPFN | 0.842 | 0.849 |

When comparing the pretraining datasets, we see that the best method differs by benchmark. Fine-tuned TabForest is the best on WhyTrees, while fine-tuned TabPFN is favored on TabZilla. As TabForest strength comes from generating complex decision boundaries, we conjecture that TabZilla has many datasets for which this property is not helpful. The zero-shot variants on both benchmarks clearly favor TabPFN, which is unsurprising given the unrealistic nature of the forest dataset generator.

The TabForestPFN combines the best of both worlds. For WhyTrees, we can see that the fine-tuned TabForestPFN has almost the same performance as TabForest, and for TabZilla, it has almost the same performance as TabPFN. We can also see that mixing in the forest dataset does not deteriorate

Table 3: Main Results on TabZilla. N. Accuracy stands for Normalized accuracy. Rank compares the relative rank of a method compared to all other methods on that dataset.

| Models | Rank | | | | N. Accuracy | |
|---|---|---|---|---|---|---|
| | min | max | mean | median | mean | median |
| TabForestPFN - Fine-tuned | 1 | 27 | **8.4** | **7.0** | 0.840 | **0.902** |
| TabPFN (retrained) - Fine-tuned | 1 | 26 | **8.4** | 7.5 | **0.843** | 0.891 |
| CatBoost | 1 | **23** | 9.6 | 9.0 | 0.842 | 0.874 |
| TabPFN (original) - Fine-tuned | 1 | 26 | 9.6 | 10.0 | 0.834 | **0.902** |
| TabForestPFN - Zero-shot | 1 | 25 | 9.7 | 9.2 | 0.819 | 0.883 |
| XGBoost | 1 | 23 | 9.8 | 9.8 | 0.836 | 0.899 |
| TabForest - Fine-tuned | 1 | 27 | 10.9 | 10.0 | 0.806 | 0.873 |
| TabPFN (retrained) - Zero-shot | 1 | 26 | 11.2 | 10.5 | 0.797 | 0.858 |
| LightGBM | 1 | 27 | 11.8 | 12.2 | 0.787 | 0.867 |
| TabPFN (original) - Zero-shot | 1 | 26 | 12.1 | 12.0 | 0.777 | 0.841 |
| RandomForest | 1 | 26 | 12.1 | 12.0 | 0.792 | 0.851 |
| Resnet | 1 | 27 | 12.8 | 12.0 | 0.727 | 0.837 |
| NODE | 1 | 27 | 12.9 | 13.5 | 0.751 | 0.833 |
| SAINT | 1 | 27 | 13.1 | 13.8 | 0.729 | 0.803 |
| SVM | 1 | 26 | 13.3 | 14.0 | 0.710 | 0.801 |
| FT-Transformer | 1 | 24 | 13.6 | 13.2 | 0.733 | 0.805 |
| TabScratch - Fine-tuned | 1 | 25 | 13.6 | 13.0 | 0.752 | 0.819 |
| DANet | 2 | 27 | 15.6 | 16.0 | 0.718 | 0.769 |
| TabForest - Zero-shot | 3 | 26 | 15.7 | 16.0 | 0.709 | 0.822 |
| MLP-rtdl | 1 | 27 | 16.9 | 19.0 | 0.619 | 0.736 |
| STG | 1 | 27 | 17.1 | 19.0 | 0.592 | 0.664 |
| LinearRegression | 1 | 27 | 18.5 | 21.0 | 0.564 | 0.593 |
| MLP | 2 | 27 | 18.8 | 21.0 | 0.570 | 0.586 |
| TabNet | 2 | 27 | 19.1 | 20.2 | 0.579 | 0.666 |
| DecisionTree | 1 | 27 | 19.7 | 21.5 | 0.502 | 0.551 |
| KNN | 2 | 27 | 20.5 | 23.0 | 0.473 | 0.478 |
| VIME | 3 | 27 | 22.9 | 25.0 | 0.343 | 0.241 |
| TabScratch - Zero-shot | 28 | 28 | 28.0 | 28.0 | 0.000 | 0.000 |

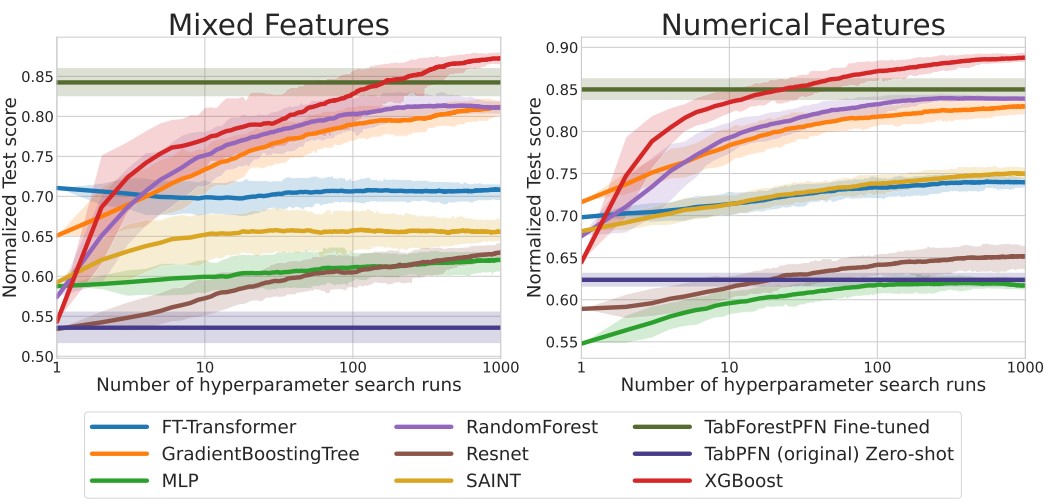

Figure 4: Main results on the WhyTrees Benchmark. TabForestPFN shows the mean over ten default runs for different fine-tuning seeds, all others use random search over the hyperparameters. See Table 2 for other ICL-transformers.

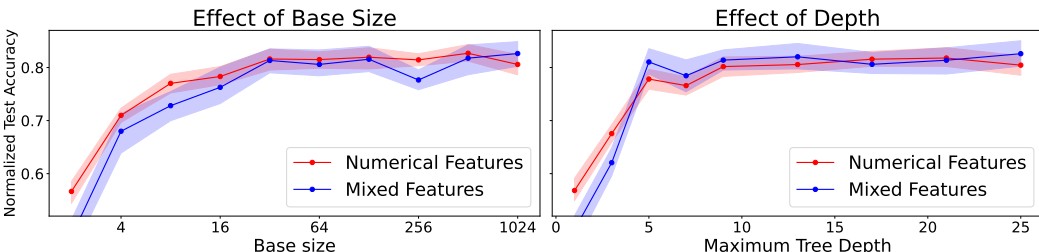

Figure 5: Ablation of the base size and maximum tree depth parameters of the Forest Dataset Generator. Figure shows normalized test accuracy of TabForest on the WhyTrees benchmark.

the zero-shot performance on either benchmark, which establishes the TabForestPFN as the clear best method among the ICL-transformer variants.

### 5.3 COMPLEXITY OF ICL-TRANSFORMERS' DECISION BOUNDARIES

In the previous section, we have seen that TabForest and TabPFN both achieve excellent performance as neural networks. Now we take a look at their decision boundaries. Repeating the analysis of the WhyTrees' authors (Grinsztajn et al., 2022), we use the Electricity dataset (OpenML ID: 44156) to predict a binary target on two features.

To capture the complexity of the decision boundary, we define the complexity score $V$. We split the feature space into a total of $n$ grid cells where each cell has a predicted probability $p_{ij}$ for grid cell indices $(i, j)$. The complexity score $V$ is defined as the sum of absolute values between neighbor cells:

$$V = \frac{1}{n} \sum_{ij} |p_{i+1,j} - p_{ij}| + |p_{i-1,j} - p_{i,j}| + |p_{i,j+1} - p_{ij}| + |p_{i,j-1} - p_{ij}|$$

The complexity score represents how fast the prediction changes when moving along the grid.

We plot the results in Figure 1. We see that when fine-tuning, both TabPFN and TabForest can create decision boundaries that are more complex than their zero-shot variants. The complexity of the decision boundaries was one of the characteristics that explained why tree-based methods outperformed neural networks (Grinsztajn et al., 2022). These results suggest ICL-transformers can also create complexity in their decision boundaries.

In our intuition, the ICL-transformer learns how to create these decision boundaries during pretraining. We can interpret this from a weight initialization perspective. The weights of the pretrained ICL-transformer provide a good initialization for the model to create complex decision boundaries, while an ICL-transformer trained from scratch lacks this ability. For this reason, TabForest can create decision boundaries of higher complexity than TabPFN.

### 5.4 ABLATION OF THE FOREST DATASET GENERATOR

In Section 4.1, we discussed two ways to influence the complexity of the forest dataset generator: the tree depth and the base size, which is the number of observations to fit the tree algorithm. We expect the performance of TabForest to increase when the complexity of the forest dataset generator increases.

In Figure 5 we show the results of pretraining different settings of base size and maximum tree depth on the WhyTrees benchmark. The tree depth is set to 1-25 as the base size changes, and the base size is fixed to 1024 as the tree depth changes. When scaling up the base size from 2 to 32 and the tree depth from 1 to 9, we observe that the performance increases, and stabilizes for higher complexities. We provide figures of the data generated with these lower complexity hyperparameters in Appendix A.10 to give an impression. The correlation between performance and complexity supports our claim that learning complex decision boundaries is the driving force behind the performance of fine-tuned TabForest.

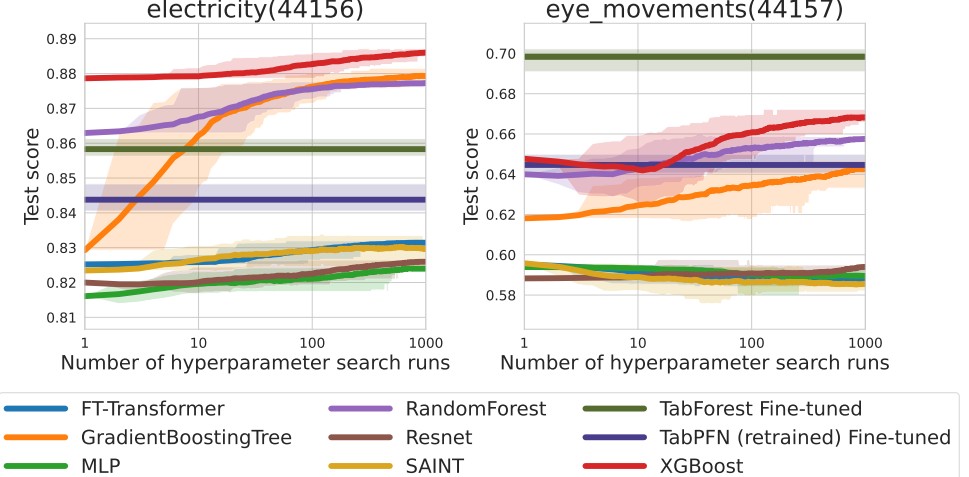

Figure 6: Comparison of fine-tuned TabForest and TabPFN on two datasets of the WhyTrees benchmark. These datasets show a big gap between neural networks and tree-based methods.

## 5.5 CASE STUDY OF THE GAP BETWEEN NEURAL NETWORKS AND TREE ALGORITHMS.

In Figure 6 we show the performance of fine-tuned TabPFN and TabForest on two specific datasets from WhyTrees. We selected these two datasets for the large gap between the neural networks (MLP, Resnet, SAINT, FT-Transformer) and the tree-based algorithms (XGBoost, GradientBoostingTree, RandomForest). The figure illustrates that ICL-transformers behave differently than other neural networks: their performance is closer to that of tree-based methods.

We propose the following explanation: these two datasets need highly complex decision boundaries. As tree-based methods are capable of creating these complex decision boundaries while neural networks struggle (Shah et al., 2020; Grinsztajn et al., 2022), fine-tuned ICL-transformers can also create them, as seen in Section 5.3. Furthermore, TabForest is naturally better at creating these decision boundaries than TabPFN, given that the forest dataset generator was specifically designed for this purpose. Figure 6 shows that TabForest significantly outperforms TabPFN on these two datasets. The explanation of the gap is further supported by Figure 1, as it illustrates the complexity of the decision boundaries for two variables from the Electricity dataset.

## 5.6 IMPROVEMENT OF FINE-TUNING OVER ZERO-SHOT

In the main results, we have seen that fine-tuning performs better than zero-shot. We look at this comparison in more detail. Figure 7a presents the performance of TabForestPFN on individual datasets from TabZilla. We can see clearly that fine-tuning strongly outperforms zero-shot when there are more than 10,000 observations. Overall, fine-tuning outperforms zero-shot on 57% of the datasets. This percentage decreases to 47% for datasets smaller than a 1000 observations and increases to 73% for datasets larger than a 1000 observations.

Figure 7b illustrates the effect of context length on the performance of TabForestPFN on the zero-shot and the fine-tuning task. We see that a higher support size is always better, which is why we set the support size in our paper to 8192, even though we only pretrained on a maximum size of 1024 observations. In conclusion, fine-tuning on the largest possible support size appears to be the most effective approach for ICL-transformers.

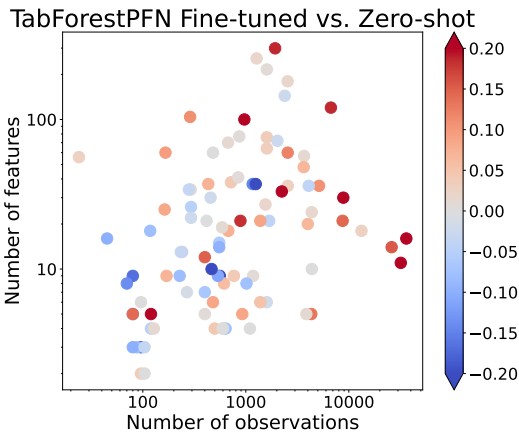

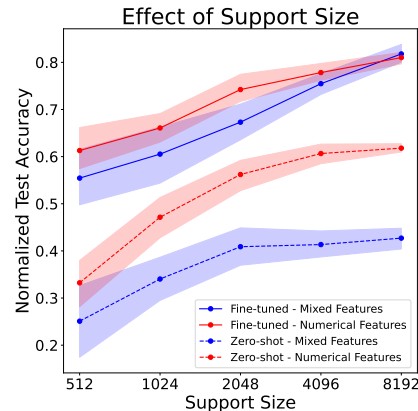

(a) Differences in normalized accuracy of individual datasets from TabZilla. The color red means fine-tuning is the best. The darkest red represents at least 0.20 normalized score points improvement, and dark blue at least 0.20 normalized accuracy points degradation.

(b) Effect of different support sizes on the WhyTrees benchmark. Both fine-tuning and zero-shot performance improves with context size.

Figure 7: Evaluation of TabForestPFN

# 6    CONCLUSION

The introduction of TabPFN (Hollmann et al., 2023) has opened up a new field of *in-context learning* (ICL)-transformers for tabular data classification. Our research has demonstrated that fine-tuned ICL-transformers achieve excellent performance and also learn to create complex decision boundaries. Furthermore, by adding the forest dataset to the pretraining mixture, we achieved performance levels competitive with tree-based methods.

Despite these advancements, there are still obstacles for ICL-transformers to overcome if we want them to replace tree-based methods in the realm of tabular data. One major challenge is the performance limitation of ICL-transformers due to GPU memory constraints. Our work uses fine-tuning as a solution to this problem, but it would be valuable to compare this approach to other concurrent research such as prompt tuning (Feuer et al., 2024), in-context distillation (Ma et al., 2023) and retrieval (Thomas et al., 2024). Moreover, our research focused solely on classification, although we expect that a simple switch from cross-entropy loss to mean-squared-error loss would suffice to tackle regression tasks. Another area that requires exploration is the setting with an exceptionally high number of features (Cherepanova et al., 2024), where the performance of ICL-transformers is unknown (McCarter, 2024). Lastly, tree-based methods can explain which features are important for their predictions, and research is needed to determine if ICL transformers can achieve a similar feat (Rundel et al., 2024). Overcoming these challenges will cement the ICL-transformer as the clear successor to tree-based methods.

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

# A APPENDIX

## A.1 ETHICS AND SOCIAL IMPACT

Improving tabular data classification can provide major benefits to society. From medical to physics applications, better performance can save lives and money. There are, however, also more nefarious applications of tabular data classification, such as fraud risk detections based on ethnics or nationality; population analysis for micro-targeting political ad campaigns; and insurance premium discrimination based on underlying medical conditions.

Our models cannot detect the purpose for which the model is used. In contrast to large language models, our tabular data models take numerical data as input. Ethnicity, gender, or other sensitive information is represented by a class number. This means our models cannot recover the meaning behind the numbers.

A big benefit of this 'numerical anonymization' is privacy. In our paper, we use synthetic data, which is completely privacy risk free. But even when pretrained on real data, recovering the original data can be extremely hard due to the lack of labels and contextual information.

In light of the above, we decide to publish the model and open access to anyone. We do not know an effective way to create any form of safeguards against misuse, and we would welcome any advice from the research community that addresses this issue.

## A.2 CODE AVAILABILITY

Code is attached to this submission.

The code includes everything: downloading datasets, preprocessing result benchmarks, training the ICL-transformers, pretrained weights on Google Drive, notebooks with analysis, and an easy example to get you started with applying the TabForestPFN to your dataset.

The code is built upon the works of Grinsztajn et al.; Hollmann et al. and McElfresh et al.. Although this resulted in a Frankenstein monster of a codebase, we have made great efforts to rewrite most of their code to integrate well. The code assumes a single server with access to 1 or more GPUs, with DistributedDataParallel used during pretraining and multiprocessing over different GPUs used during inference.

As currently there is no good tabular code base out there, we recommend anyone that is interested in doing research in tabular data to take a look at ours.

## A.3 DATA PREPROCESSING

Before data is put into the neural network, the data is preprocessed. We use the exact same routine for both synthetic data and real-world data to ensure minimal differences in distribution and summary statistics of the input to the transformer. Algorithm 2 presents the procedure for preprocessing.

---

**Algorithm 2** Data Preprocessing

**Require:** $X_{raw}, y_{raw}$
1: **Impute** NaN features with column mean.
2: **Remove** features with one unique value.
3: **Select** a subset of a hundred features.
4: **Transform** all features to normal using quantile transformation.
5: **Normalize** data to unit mean and variance.
6: **Scale** data based on number of features.
7: **Pad** the features to $d_f$ features by adding zeros.
8: **if** Pretraining **then**
9:     **Shuffle** the order of the features and classes.
10: **end if**
**Ensure:** $X_{preprocessed}, y_{preprocessed}$

---

Because we fixed the input size of the neural network to $d_f = 100$ features, we first select a subset of a $d_f$ features using scikit-learn's *SelectKBest* Pedregosa et al. (2011). If there are less than $d_f$ features, we add zeros to ensure exactly $d_f$ features. Following TabPFN, we scale by multiplying by $100/d_{f*}$, where $d_{f*}$ is the number of features after selecting a subset. To be robust to skewness and outliers, we transform the data using scikit-learn's *QuantileTransformer* to follow a normal distribution. We make no distinction between numerical and categorical values in all our preprocessing.

In comparison to TabPFN, our data preprocessing follows roughly the same scheme. One change is the use of a quantile transformer, while they use standard input or a power transformer. We consider this a preference, both seem to work fine.

Furthermore, the TabPFN authors like to ensemble outputs of the TabPFN architecture, varying the transformation function between standard input and power transformer. In our paper, we use no ensembling at all.

## A.4 TRAINING SETTINGS

All ICL-transformer architectures, including the original TabPFN, use the same model. The model consists of 12 layers, 4 attention heads, a hidden dimension of 512, and 10 classes as output dimension. Pretraining uses batch size 512, learning rate 1e-4, weight decay 0.00, AdamW optimizer with betas (0.9, 0.95), cosine scheduling, and maximum global gradient norm 1.0. Fine-tuning is performed under batch size 1, learning rate 1e-5, weight decay 0.00, no scheduling, with early stopping, and a maximum of 300 steps. To fit the model on an RTX 3090 with 24GB during fine-tuning, we set the maximum support size to 8192 samples and the maximum query size to 1024. Pre-training uses data generated with maximum support size 1024 and maximum query size 128. We choose these settings because the maximum support size affects performance, but the maximum query size only affects inference speed. On these settings, we train all ICL-transformers for 50,000 steps, which takes 4 GPU-days on one H100. Running all benchmarks takes one additional H100 GPU-day.

## A.5 BENCHMARK METADATA

Of both the TabZilla and the WhyTrees benchmark, we show the OpenML Vanschoren et al. (2014) datasets we use as well as their characteristics. See Table 4 and Table 5. The TabZilla table presents the 94 datasets picked out of the 176 total datasets.

From the original 176 Tabzilla datasets, we excluded every dataset that does not have at least one completed run on default settings for every model, which brings the value to 99. Additionally, we exclude four datasets because they have more than 10 classes. The preprocessing code of one other dataset did not run without errors, and so is removed as well. The TabZilla authors did experiment with running TabPFN, but only on 62 datasets with a maximum support size of 3000 samples, so we redo their experiment.

Table 4: Metadata of the WhyTrees Benchmark. Splits refers to the number of cross validation splits.

| OpenML | | Observations | | | | Features | Splits | Classes |
|---|---|---|---|---|---|---|---|---|
| ID | Name | All | Train | Valid | Test | | | |
| 44089 | credit | 16714 | 10000 | 2014 | 4700 | 10 | 2 | 2 |
| 44120 | electricity | 38474 | 10000 | 8542 | 19932 | 7 | 1 | 2 |
| 44121 | covertype | 566602 | 10000 | 50000 | 50000 | 10 | 1 | 2 |
| 44122 | pol | 10082 | 7057 | 907 | 2118 | 26 | 3 | 2 |
| 44123 | house_16H | 13488 | 9441 | 1214 | 2833 | 16 | 3 | 2 |
| 44125 | MagicTelescope | 13376 | 9363 | 1203 | 2810 | 10 | 3 | 2 |
| 44126 | bank-marketing | 10578 | 7404 | 952 | 2222 | 7 | 3 | 2 |
| 44128 | MiniBooNE | 72998 | 10000 | 18899 | 44099 | 50 | 1 | 2 |
| 44129 | Higgs | 940160 | 10000 | 50000 | 50000 | 24 | 1 | 2 |
| 44130 | eye_movements | 7608 | 5325 | 684 | 1599 | 20 | 3 | 2 |
| 44156 | electricity | 38474 | 10000 | 8542 | 19932 | 8 | 1 | 2 |
| 44157 | eye_movements | 7608 | 5325 | 684 | 1599 | 23 | 3 | 2 |

| 44159 | covertype | 423680 | 10000 | 50000 | 50000 | 54 | 1 | 2 |
| 45019 | Bioresponse | 3434 | 2403 | 309 | 722 | 419 | 5 | 2 |
| 45020 | default-of-cred... | 13272 | 9290 | 1194 | 2788 | 20 | 3 | 2 |
| 45021 | jannis | 57580 | 10000 | 14274 | 33306 | 54 | 1 | 2 |
| 45022 | Diabetes130US | 71090 | 10000 | 18327 | 42763 | 7 | 1 | 2 |
| 45026 | heloc | 10000 | 7000 | 900 | 2100 | 22 | 3 | 2 |
| 45028 | california | 20634 | 10000 | 3190 | 7444 | 8 | 1 | 2 |
| 45035 | albert | 58252 | 10000 | 14475 | 33777 | 31 | 1 | 2 |
| 45036 | default-of-cred... | 13272 | 9290 | 1194 | 2788 | 21 | 3 | 2 |
| 45038 | road-safety | 111762 | 10000 | 30528 | 50000 | 32 | 1 | 2 |
| 45039 | compas-two-year... | 4966 | 3476 | 447 | 1043 | 11 | 3 | 2 |

Table 5: Metadata of the TabZilla Benchmark. Splits refers to the number of cross validation splits.

| OpenML | | Observations | | | | Features | Splits | Classes |
|---|---|---|---|---|---|---|---|---|
| ID | Name | All | Train | Valid | Test | | | |
| 3 | kr-vs-kp | 3196 | 2556 | 320 | 320 | 36 | 10 | 2 |
| 4 | labor | 57 | 45 | 6 | 6 | 16 | 10 | 2 |
| 9 | autos | 205 | 163 | 21 | 21 | 25 | 10 | 6 |
| 10 | lymph | 148 | 118 | 15 | 15 | 18 | 10 | 4 |
| 11 | balance-scale | 625 | 499 | 63 | 63 | 4 | 10 | 3 |
| 12 | mfeat-factors | 2000 | 1600 | 200 | 200 | 216 | 10 | 10 |
| 14 | mfeat-fourier | 2000 | 1600 | 200 | 200 | 76 | 10 | 10 |
| 15 | breast-w | 699 | 559 | 70 | 70 | 9 | 10 | 2 |
| 16 | mfeat-karhunen | 2000 | 1600 | 200 | 200 | 64 | 10 | 10 |
| 18 | mfeat-morpholog... | 2000 | 1600 | 200 | 200 | 6 | 10 | 10 |
| 23 | cmc | 1473 | 1177 | 148 | 148 | 9 | 10 | 3 |
| 25 | colic | 368 | 294 | 37 | 37 | 26 | 10 | 2 |
| 27 | colic | 368 | 294 | 37 | 37 | 22 | 10 | 2 |
| 29 | credit-approval | 690 | 552 | 69 | 69 | 15 | 10 | 2 |
| 30 | page-blocks | 5473 | 4377 | 548 | 548 | 10 | 10 | 5 |
| 35 | dermatology | 366 | 292 | 37 | 37 | 34 | 10 | 6 |
| 37 | diabetes | 768 | 614 | 77 | 77 | 8 | 10 | 2 |
| 39 | sonar | 208 | 166 | 21 | 21 | 60 | 10 | 2 |
| 40 | glass | 214 | 170 | 22 | 22 | 9 | 10 | 6 |
| 43 | spambase | 4601 | 3680 | 460 | 461 | 57 | 10 | 2 |
| 45 | splice | 3190 | 2552 | 319 | 319 | 60 | 10 | 3 |
| 47 | tae | 151 | 120 | 15 | 16 | 5 | 10 | 3 |
| 48 | heart-c | 303 | 241 | 31 | 31 | 13 | 10 | 2 |
| 49 | tic-tac-toe | 958 | 766 | 96 | 96 | 9 | 10 | 2 |
| 50 | heart-h | 294 | 234 | 30 | 30 | 13 | 10 | 2 |
| 53 | vehicle | 846 | 676 | 85 | 85 | 18 | 10 | 4 |
| 59 | iris | 150 | 120 | 15 | 15 | 4 | 10 | 3 |
| 2074 | satimage | 6430 | 5144 | 643 | 643 | 36 | 10 | 6 |
| 2079 | eucalyptus | 736 | 588 | 74 | 74 | 19 | 10 | 5 |
| 2867 | anneal | 898 | 718 | 90 | 90 | 38 | 10 | 5 |
| 3485 | scene | 2407 | 1925 | 241 | 241 | 299 | 10 | 2 |
| 3512 | synthetic_contr... | 600 | 480 | 60 | 60 | 60 | 10 | 6 |
| 3540 | analcatdata_box... | 120 | 96 | 12 | 12 | 3 | 10 | 2 |
| 3543 | irish | 500 | 400 | 50 | 50 | 5 | 10 | 2 |
| 3549 | analcatdata_aut... | 841 | 672 | 84 | 85 | 70 | 10 | 4 |
| 3560 | analcatdata_dmf... | 797 | 637 | 80 | 80 | 4 | 10 | 6 |
| 3561 | profb | 672 | 536 | 68 | 68 | 9 | 10 | 2 |
| 3602 | visualizing_env... | 111 | 88 | 11 | 12 | 3 | 10 | 2 |
| 3620 | fri_c0_100_5 | 100 | 80 | 10 | 10 | 5 | 10 | 2 |
| 3647 | rabe_266 | 120 | 96 | 12 | 12 | 2 | 10 | 2 |
| 3711 | elevators | 16599 | 13279 | 1660 | 1660 | 18 | 10 | 2 |
| 3731 | visualizing_liv... | 130 | 104 | 13 | 13 | 2 | 10 | 2 |
| 3739 | analcatdata_chl... | 100 | 80 | 10 | 10 | 3 | 10 | 2 |
| 3748 | transplant | 131 | 104 | 13 | 14 | 3 | 10 | 2 |
| 3779 | fri_c3_100_5 | 100 | 80 | 10 | 10 | 5 | 10 | 2 |

| 3797 | socmob | 1156 | 924 | 116 | 116 | 5 | 10 | 2 |
|---|---|---|---|---|---|---|---|---|
| 3896 | ada_agnostic | 4562 | 3648 | 457 | 457 | 48 | 10 | 2 |
| 3902 | pc4 | 1458 | 1166 | 146 | 146 | 37 | 10 | 2 |
| 3903 | pc3 | 1563 | 1249 | 157 | 157 | 37 | 10 | 2 |
| 3904 | jm1 | 10885 | 8707 | 1089 | 1089 | 21 | 10 | 2 |
| 3913 | kc2 | 522 | 416 | 53 | 53 | 21 | 10 | 2 |
| 3917 | kc1 | 2109 | 1687 | 211 | 211 | 21 | 10 | 2 |
| 3918 | pc1 | 1109 | 887 | 111 | 111 | 21 | 10 | 2 |
| 3953 | adult-census | 32561 | 26048 | 3256 | 3257 | 14 | 10 | 2 |
| 9946 | wdbc | 569 | 455 | 57 | 57 | 30 | 10 | 2 |
| 9952 | phoneme | 5404 | 4322 | 541 | 541 | 5 | 10 | 2 |
| 9957 | qsar-biodeg | 1055 | 843 | 106 | 106 | 41 | 10 | 2 |
| 9960 | wall-robot-navi... | 5456 | 4364 | 546 | 546 | 24 | 10 | 4 |
| 9964 | semeion | 1593 | 1273 | 160 | 160 | 256 | 10 | 10 |
| 9971 | ilpd | 583 | 465 | 59 | 59 | 10 | 10 | 2 |
| 9978 | ozone-level-8hr | 2534 | 2026 | 254 | 254 | 72 | 10 | 2 |
| 9984 | fertility | 100 | 80 | 10 | 10 | 9 | 10 | 2 |
| 10089 | acute-inflammat... | 120 | 96 | 12 | 12 | 6 | 10 | 2 |
| 10093 | banknote-authen... | 1372 | 1096 | 138 | 138 | 4 | 10 | 2 |
| 10101 | blood-transfusi... | 748 | 598 | 75 | 75 | 4 | 10 | 2 |
| 14952 | PhishingWebsite... | 11055 | 8843 | 1106 | 1106 | 30 | 10 | 2 |
| 14954 | cylinder-bands | 540 | 432 | 54 | 54 | 37 | 10 | 2 |
| 14965 | bank-marketing | 45211 | 36168 | 4521 | 4522 | 16 | 10 | 2 |
| 14967 | cjs | 2796 | 2236 | 280 | 280 | 33 | 10 | 6 |
| 125920 | dresses-sales | 500 | 400 | 50 | 50 | 12 | 10 | 2 |
| 125921 | LED-display-dom... | 500 | 400 | 50 | 50 | 7 | 10 | 10 |
| 145793 | yeast | 1269 | 1015 | 127 | 127 | 8 | 10 | 4 |
| 145799 | breast-cancer | 286 | 228 | 29 | 29 | 9 | 10 | 2 |
| 145836 | blood-transfusi... | 748 | 598 | 75 | 75 | 4 | 10 | 2 |
| 145847 | hill-valley | 1212 | 968 | 122 | 122 | 100 | 10 | 2 |
| 145977 | ecoli | 336 | 268 | 34 | 34 | 7 | 10 | 8 |
| 145984 | ionosphere | 351 | 280 | 35 | 36 | 34 | 10 | 2 |
| 146024 | lung-cancer | 32 | 24 | 4 | 4 | 56 | 10 | 3 |
| 146063 | hayes-roth | 160 | 128 | 16 | 16 | 4 | 10 | 3 |
| 146065 | monks-problems-... | 601 | 480 | 60 | 61 | 6 | 10 | 2 |
| 146192 | car-evaluation | 1728 | 1382 | 173 | 173 | 21 | 10 | 4 |
| 146210 | postoperative-p... | 88 | 70 | 9 | 9 | 8 | 10 | 2 |
| 146607 | SpeedDating | 8378 | 6702 | 838 | 838 | 120 | 10 | 2 |
| 146800 | MiceProtein | 1080 | 864 | 108 | 108 | 77 | 10 | 8 |
| 146817 | steel-plates-fa... | 1941 | 1552 | 194 | 195 | 27 | 10 | 7 |
| 146818 | Australian | 690 | 552 | 69 | 69 | 14 | 10 | 2 |
| 146820 | wilt | 4839 | 3871 | 484 | 484 | 5 | 10 | 2 |
| 146821 | car | 1728 | 1382 | 173 | 173 | 6 | 10 | 4 |
| 167140 | dna | 3186 | 2548 | 319 | 319 | 180 | 10 | 3 |
| 167141 | churn | 5000 | 4000 | 500 | 500 | 20 | 10 | 2 |
| 167211 | Satellite | 5100 | 4080 | 510 | 510 | 36 | 10 | 2 |
| 168911 | jasmine | 2984 | 2386 | 299 | 299 | 144 | 10 | 2 |
| 190408 | Click_predictio... | 39948 | 31958 | 3995 | 3995 | 11 | 10 | 2 |
| 360948 | libras | 360 | 288 | 36 | 36 | 104 | 10 | 10 |

## A.6 RUN TIMES

Table 6 and 7 present the run times of TabForestPFN on both the WhyTrees and the TabZilla benchmark. All fine-tuning runs take at most 220 seconds per cross validation split, with an average of 68 seconds. Runtimes differ by GPU, and creating a fair comparison with CPU-based methods is difficult due to the different hardware used. As main takeaway, we recommend to summarize the fine-tuning run time as "a few minutes at most for a dataset of around 10,000 observations".

Table 6: Run times of TabForestPFN of the WhyTrees Benchmark. The runtime is the end-to-end time in seconds for one cross validation split. End-to-end time includes loading, preprocessing, training and testing.

| OpenML | | Size | | Run time (s) | |
|---|---|---|---|---|---|
| ID | Name | Obs. | Feat. | Zero-shot | Fine-tuned |
| 44089 | credit | 10000 | 10 | 9 | 103 |
| 44120 | electricity | 10000 | 7 | 15 | 151 |
| 44121 | covertype | 10000 | 10 | 34 | 167 |
| 44122 | pol | 7057 | 26 | 6 | 57 |
| 44123 | house_16H | 9441 | 16 | 8 | 72 |
| 44125 | MagicTelescope | 9363 | 10 | 7 | 105 |
| 44126 | bank-marketing | 7404 | 7 | 7 | 68 |
| 44128 | MiniBooNE | 10000 | 50 | 28 | 126 |
| 44129 | Higgs | 10000 | 24 | 34 | 119 |
| 44130 | eye_movements | 5325 | 20 | 5 | 63 |
| 44156 | electricity | 10000 | 8 | 17 | 142 |
| 44157 | eye_movements | 5325 | 23 | 6 | 65 |
| 44159 | covertype | 10000 | 54 | 37 | 219 |
| 45019 | Bioresponse | 2403 | 419 | 8 | 34 |
| 45020 | default-of-credit-card-clients | 9290 | 20 | 7 | 81 |
| 45021 | jannis | 10000 | 54 | 23 | 130 |
| 45022 | Diabetes130US | 10000 | 7 | 25 | 95 |
| 45026 | heloc | 7000 | 22 | 6 | 56 |
| 45028 | california | 10000 | 8 | 11 | 112 |
| 45035 | albert | 10000 | 31 | 21 | 103 |
| 45036 | default-of-credit-card-clients | 9290 | 21 | 8 | 79 |
| 45038 | road-safety | 10000 | 32 | 30 | 153 |
| 45039 | compas-two-years | 3476 | 11 | 5 | 43 |

Table 7: Run times of TabForestPFN of the TabZilla Benchmark. The runtime is the end-to-end time in seconds for one cross validation split. End-to-end time includes loading, preprocessing, training and testing.

| OpenML | | Size | | Run time (s) | |
|---|---|---|---|---|---|
| ID | Name | Obs. | Feat. | Zero-shot | Fine-tuned |
| 3 | kr-vs-kp | 2556 | 36 | 4 | 29 |
| 4 | labor | 45 | 16 | 3 | 13 |
| 9 | autos | 163 | 25 | 3 | 11 |
| 10 | lymph | 118 | 18 | 3 | 9 |
| 11 | balance-scale | 499 | 4 | 3 | 32 |
| 12 | mfeat-factors | 1600 | 216 | 5 | 26 |
| 14 | mfeat-fourier | 1600 | 76 | 4 | 29 |
| 15 | breast-w | 559 | 9 | 3 | 19 |
| 16 | mfeat-karhunen | 1600 | 64 | 4 | 22 |
| 18 | mfeat-morphological | 1600 | 6 | 4 | 22 |
| 23 | cmc | 1177 | 9 | 4 | 20 |
| 25 | colic | 294 | 26 | 3 | 10 |
| 27 | colic | 294 | 22 | 3 | 11 |
| 29 | credit-approval | 552 | 15 | 4 | 22 |
| 30 | page-blocks | 4377 | 10 | 5 | 40 |
| 35 | dermatology | 292 | 34 | 3 | 13 |
| 37 | diabetes | 614 | 8 | 3 | 19 |
| 39 | sonar | 166 | 60 | 3 | 11 |
| 40 | glass | 170 | 9 | 3 | 10 |
| 43 | spambase | 3680 | 57 | 6 | 42 |
| 45 | splice | 2552 | 60 | 3 | 33 |
| 47 | tae | 120 | 5 | 3 | 11 |
| 48 | heart-c | 241 | 13 | 3 | 11 |

| 49 | tic-tac-toe | 766 | 9 | 3 | 20 |
|---|---|---|---|---|---|
| 50 | heart-h | 234 | 13 | 2 | 12 |
| 53 | vehicle | 676 | 18 | 3 | 23 |
| 59 | iris | 120 | 4 | 3 | 16 |
| 2074 | satimage | 5144 | 36 | 6 | 55 |
| 2079 | eucalyptus | 588 | 19 | 3 | 18 |
| 2867 | anneal | 718 | 38 | 3 | 26 |
| 3485 | scene | 1925 | 299 | 6 | 37 |
| 3512 | synthetic_control | 480 | 60 | 3 | 18 |
| 3540 | analcatdata_boxing1 | 96 | 3 | 3 | 12 |
| 3543 | irish | 400 | 5 | 4 | 19 |
| 3549 | analcatdata_authorship | 672 | 70 | 4 | 25 |
| 3560 | analcatdata_dmft | 637 | 4 | 3 | 20 |
| 3561 | profb | 536 | 9 | 3 | 16 |
| 3602 | visualizing_environmental | 88 | 3 | 3 | 10 |
| 3620 | fri_c0_100_5 | 80 | 5 | 3 | 13 |
| 3647 | rabe_266 | 96 | 2 | 3 | 14 |
| 3711 | elevators | 13279 | 18 | 9 | 101 |
| 3731 | visualizing_livestock | 104 | 2 | 3 | 15 |
| 3739 | analcatdata_chlamydia | 80 | 3 | 3 | 16 |
| 3748 | transplant | 104 | 3 | 3 | 12 |
| 3779 | fri_c3_100_5 | 80 | 5 | 3 | 13 |
| 3797 | socmob | 924 | 5 | 3 | 19 |
| 3896 | ada_agnostic | 3648 | 48 | 6 | 36 |
| 3902 | pc4 | 1166 | 37 | 4 | 23 |
| 3903 | pc3 | 1249 | 37 | 4 | 26 |
| 3904 | jm1 | 8707 | 21 | 6 | 117 |
| 3913 | kc2 | 416 | 21 | 3 | 26 |
| 3917 | kc1 | 1687 | 21 | 4 | 48 |
| 3918 | pc1 | 887 | 21 | 3 | 16 |
| 3953 | adult-census | 26048 | 14 | 14 | 175 |
| 9946 | wdbc | 455 | 30 | 4 | 17 |
| 9952 | phoneme | 4322 | 5 | 4 | 44 |
| 9957 | qsar-biodeg | 843 | 41 | 4 | 23 |
| 9960 | wall-robot-navigation | 4364 | 24 | 5 | 42 |
| 9964 | semeion | 1273 | 256 | 5 | 26 |
| 9971 | ilpd | 465 | 10 | 4 | 20 |
| 9978 | ozone-level-8hr | 2026 | 72 | 4 | 25 |
| 9984 | fertility | 80 | 9 | 3 | 13 |
| 10089 | acute-inflammations | 96 | 6 | 3 | 10 |
| 10093 | banknote-authentication | 1096 | 4 | 4 | 20 |
| 10101 | blood-transfusion-service-center | 598 | 4 | 3 | 20 |
| 14952 | PhishingWebsites | 8843 | 30 | 8 | 103 |
| 14954 | cylinder-bands | 432 | 37 | 4 | 16 |
| 14965 | bank-marketing | 36168 | 16 | 17 | 165 |
| 14967 | cjs | 2236 | 33 | 4 | 79 |
| 125920 | dresses-sales | 400 | 12 | 4 | 18 |
| 125921 | LED-display-domain-7digit | 400 | 7 | 4 | 16 |
| 145793 | yeast | 1015 | 8 | 4 | 19 |
| 145799 | breast-cancer | 228 | 9 | 3 | 11 |
| 145836 | blood-transfusion-service-center | 598 | 4 | 3 | 21 |
| 145847 | hill-valley | 968 | 100 | 4 | 47 |
| 145977 | ecoli | 268 | 7 | 3 | 12 |
| 145984 | ionosphere | 280 | 34 | 3 | 12 |
| 146024 | lung-cancer | 24 | 56 | 3 | 14 |
| 146063 | hayes-roth | 128 | 4 | 3 | 14 |
| 146065 | monks-problems-2 | 480 | 6 | 2 | 22 |
| 146192 | car-evaluation | 1382 | 21 | 4 | 27 |
| 146210 | postoperative-patient-data | 70 | 8 | 3 | 13 |
| 146607 | SpeedDating | 6702 | 120 | 6 | 57 |
| 146800 | MiceProtein | 864 | 77 | 4 | 28 |
| 146817 | steel-plates-fault | 1552 | 27 | 4 | 22 |
| 146818 | Australian | 552 | 14 | 4 | 23 |
| 146820 | wilt | 3871 | 5 | 4 | 30 |
| 146821 | car | 1382 | 6 | 4 | 30 |

| 167140 | dna | 2548 | 180 | 4 | 26 |
| 167141 | churn | 4000 | 20 | 5 | 41 |
| 167211 | Satellite | 4080 | 36 | 5 | 40 |
| 168911 | jasmine | 2386 | 144 | 4 | 36 |
| 190408 | Click_prediction_small | 31958 | 11 | 14 | 129 |
| 360948 | libras | 288 | 104 | 3 | 11 |

## A.7 TabZilla Further Results

In the TabZilla main results Table 3, we have shown the performance including all methods implemented by the TabZilla authors. Because the rank is calculated over all included methods, which ICL-transformer variants we include might change the results. Therefore, we check if the results are the same if we use calculate the rankings one ICL-transformer at the time.

Table 8 shows the results of only the fine-tuned TabForestPFN versus the rest of the benchmark. We do this for every ICL-transformer and aggregregate the results in Table 9. All results are qualitatively the same as in Table 3.

Table 8: Main Results on TabZilla. N. Accuracy stands for Normalized accuracy. Rank compares the relative rank of a method compared to all other methods on that dataset.

| Models | Rank | | | | N. Accuracy | |
| --- | --- | --- | --- | --- | --- | --- |
| | min | max | mean | median | mean | median |
| TabForestPFN - Fine-tuned | 1 | 19 | **5.6** | **4.5** | 0.846 | **0.910** |
| CatBoost | 1 | **15** | 6.2 | 5.0 | **0.848** | 0.876 |
| XGBoost | 1 | 16 | 6.3 | 5.0 | 0.841 | 0.901 |
| LightGBM | 1 | 19 | 7.7 | 6.0 | 0.792 | 0.871 |
| RandomForest | 1 | 18 | 7.9 | 8.0 | 0.797 | 0.852 |
| NODE | 1 | 19 | 8.4 | 8.0 | 0.754 | 0.839 |
| Resnet | 1 | 19 | 8.4 | 8.0 | 0.729 | 0.837 |
| SAINT | 1 | 19 | 8.5 | 8.0 | 0.733 | 0.817 |
| SVM | 1 | 18 | 8.6 | 8.0 | 0.713 | 0.801 |
| FT-Transformer | 1 | 16 | 8.9 | 8.5 | 0.737 | 0.805 |
| DANet | 2 | 19 | 10.0 | 10.0 | 0.721 | 0.768 |
| MLP-rtdl | 1 | 19 | 11.5 | 12.0 | 0.622 | 0.737 |
| STG | 1 | 19 | 11.5 | 12.0 | 0.594 | 0.672 |
| LinearRegression | 1 | 19 | 12.3 | 13.8 | 0.567 | 0.592 |
| MLP | 1 | 19 | 12.6 | 14.0 | 0.572 | 0.588 |
| TabNet | 2 | 19 | 12.8 | 13.2 | 0.583 | 0.672 |
| DecisionTree | 1 | 19 | 13.3 | 14.0 | 0.504 | 0.551 |
| KNN | 2 | 19 | 13.9 | 15.0 | 0.475 | 0.484 |
| VIME | 1 | 19 | 15.7 | 17.0 | 0.345 | 0.240 |

## A.8 WhyTrees Further Results

The main results in Figure 4 report the normalized accuracy aggregated over all datasets. In Figure 8 and 9 we show the comparison between fine-tuned TabForestPFN and the original zero-shot version of TabPFN on all 23 datasets. In Figure 10 and 11 we show the same graphs but with fine-tuned TabPFN and fine-tuned TabForest.

## A.9 One-by-One Comparisons

In Figure 12 we plot one-to-one comparisons of fine-tuned TabForestPFN versus CatBoost, TabForest and TabPFN. We see no clear correlations in the other comparisons between performance difference and model.

Table 9: Main Results on TabZilla. N. Accuracy stands for Normalized accuracy. Rank compares the relative rank of a method compared to all other methods on that dataset. This table displays individual results: Table 8 is run individually for all ICL-transformer variants, and the row of the ICL-transformer is copy-pasted here.

| Models | Rank | | | | N. Accuracy | |
|---|---|---|---|---|---|---|
| | min | max | mean | median | mean | median |
| Zero-shot | | | | | | |
| TabScratch | 19 | 19 | 19.0 | 19.0 | 0.000 | 0.000 |
| TabPFN (original) | 1 | 19 | 7.7 | 7.0 | 0.780 | 0.841 |
| TabPFN (retrained) | 1 | 18 | 7.0 | **6.0** | 0.803 | 0.860 |
| TabForest | 1 | 19 | 9.8 | 10.0 | 0.714 | 0.822 |
| TabForestPFN | 1 | 18 | **6.3** | **6.0** | **0.824** | **0.900** |
| Fine-tuned | | | | | | |
| TabScratch | 1 | 18 | 8.3 | 7.0 | 0.757 | 0.819 |
| TabPFN (original) | 1 | 18 | 6.4 | 6.5 | 0.838 | 0.909 |
| TabPFN (retrained) | 1 | 18 | **5.6** | 5.0 | **0.847** | 0.891 |
| TabForest | 1 | 19 | 7.0 | 6.0 | 0.810 | 0.885 |
| TabForestPFN | 1 | 19 | **5.6** | **4.5** | 0.846 | **0.910** |

## A.10 SYNTHETIC DATA WITH LOWER COMPLEXITY

In the ablation we have seen that even with a forest dataset generator with lower complexity parameters, we still have similar performance. To give an idea of how complex the data is, here we showcase the generated data. Figure 13 displays generated data with base size 32, and Figure 14 displays generated data with maximum tree depth 9.

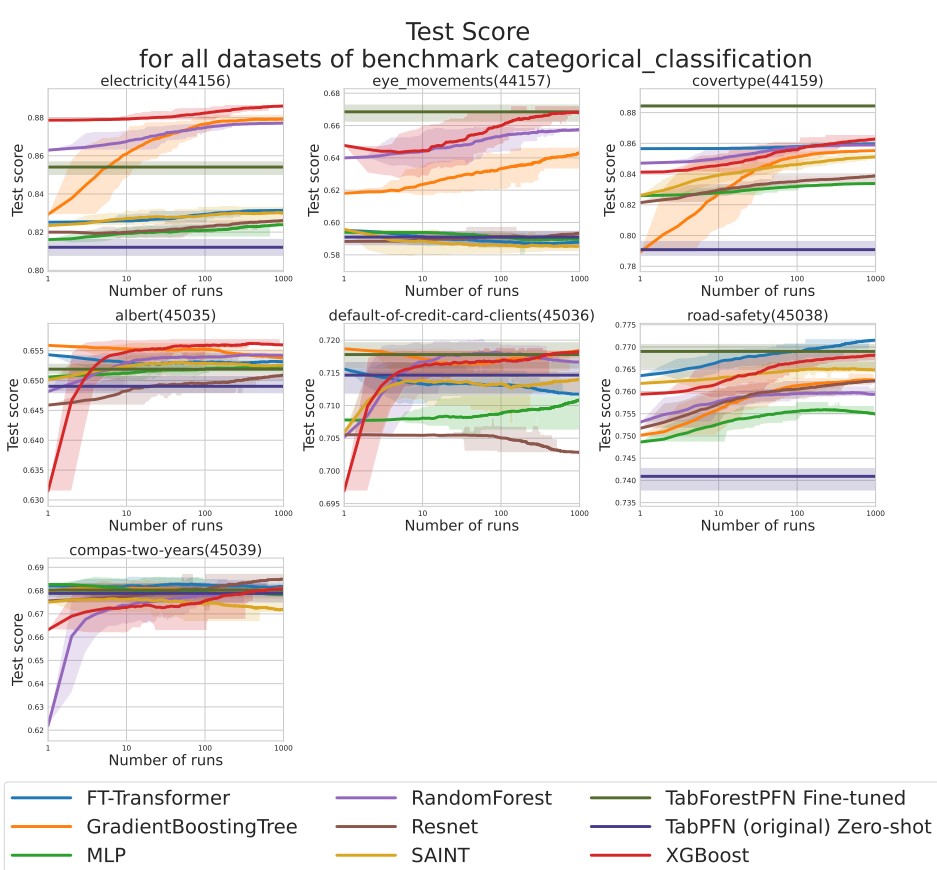

Figure 8: Comparison of fine-tuned TabForestPFN and the original zero-shot TabPFN on the WhyTrees benchmark with mixed features.

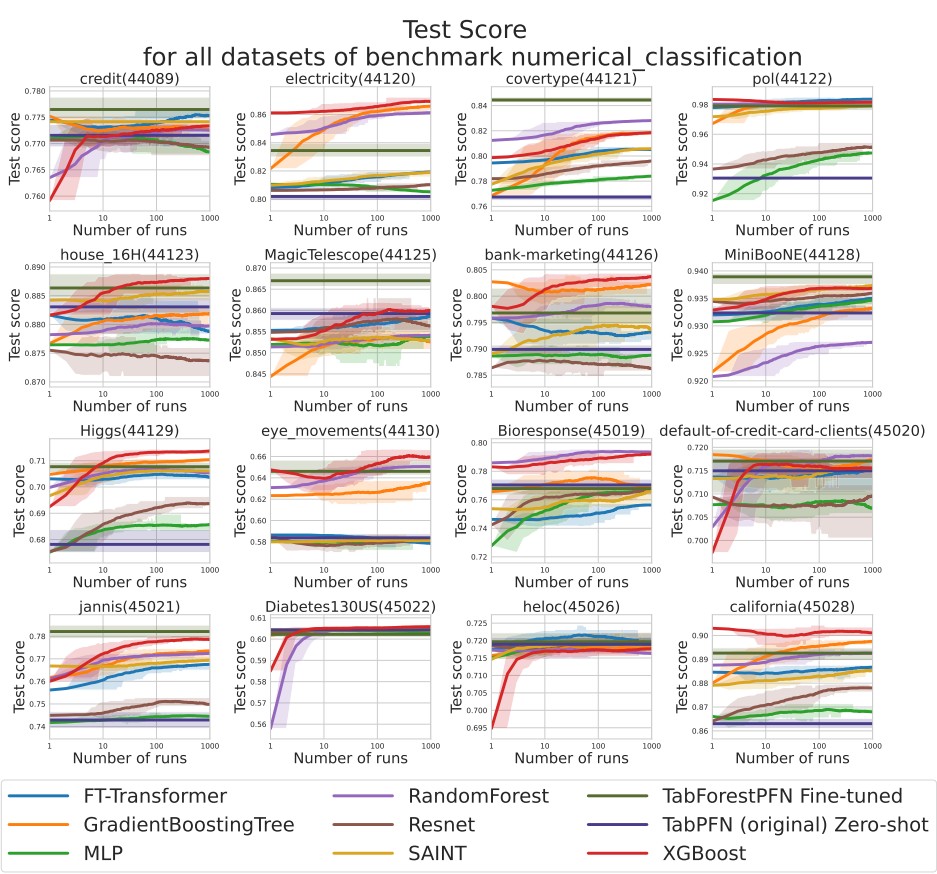

Figure 9: Comparison of fine-tuned TabForestPFN and the original zero-shot TabPFN on the WhyTrees benchmark with mixed features.

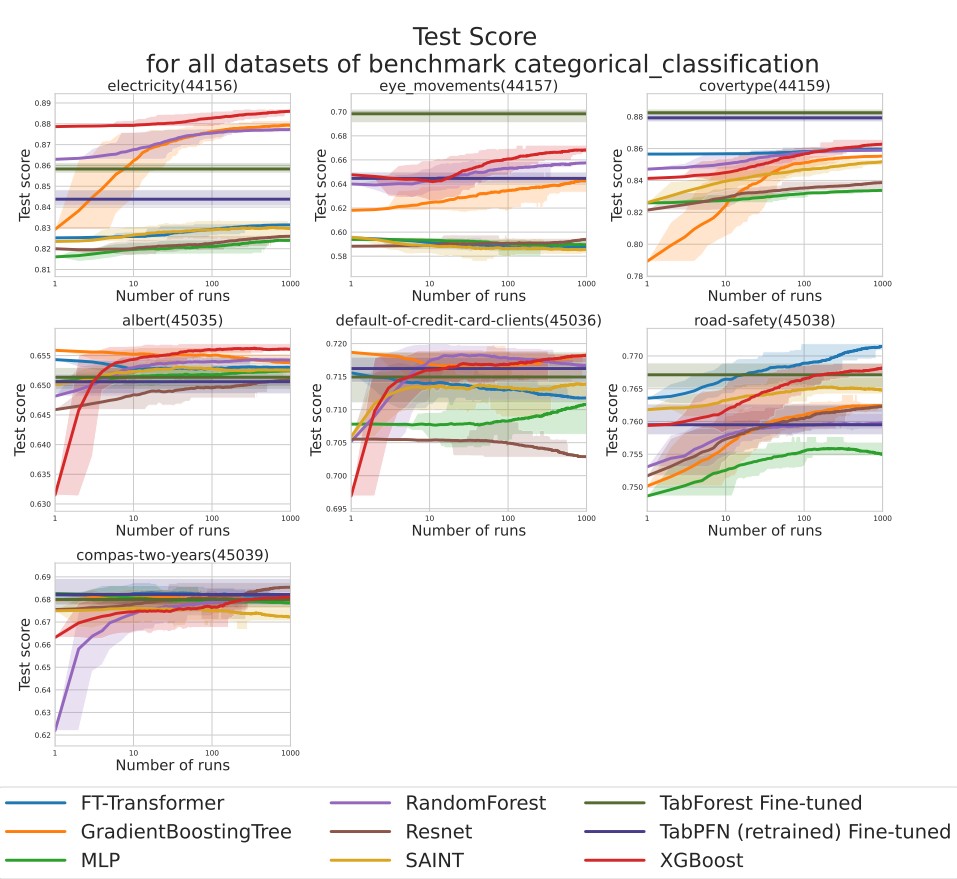

Figure 10: Comparison of fine-tuned TabForest and fine-tuned TabPFN on the WhyTrees benchmark with mixed features.

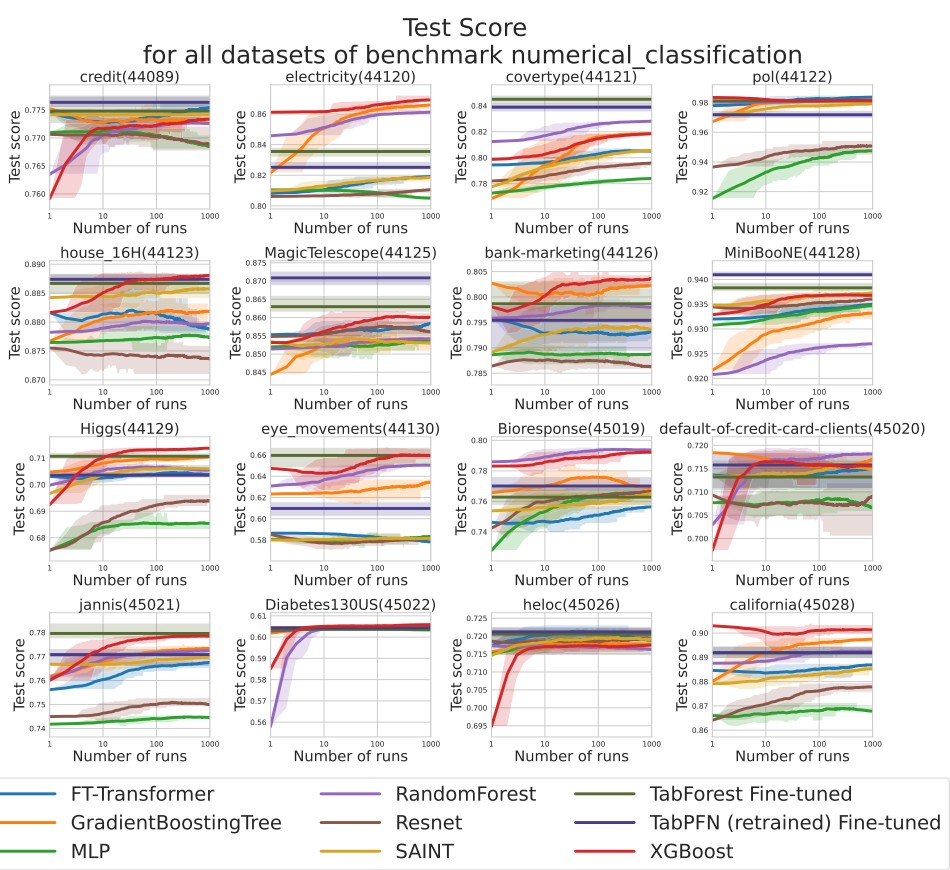

Figure 11: Comparison of fine-tuned TabForest and fine-tuned TabPFN on the WhyTrees benchmark with mixed features.

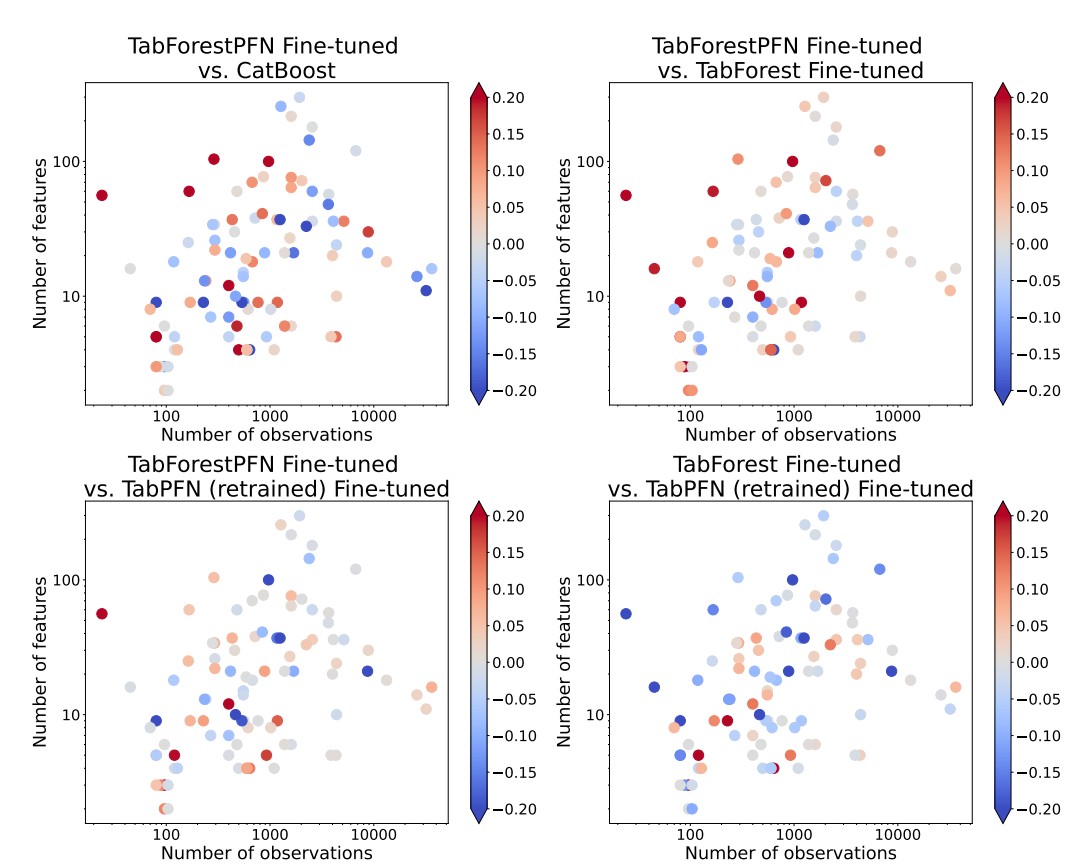

Figure 12: Differences in normalized accuracy of individual datasets from TabZilla. The color red means the left-mentioned method is the best, blue for the right-mentioned method. The darkest red represents at least 0.20 normalized score points improvement, and dark blue at least 0.20 normalized accuracy points degradation.

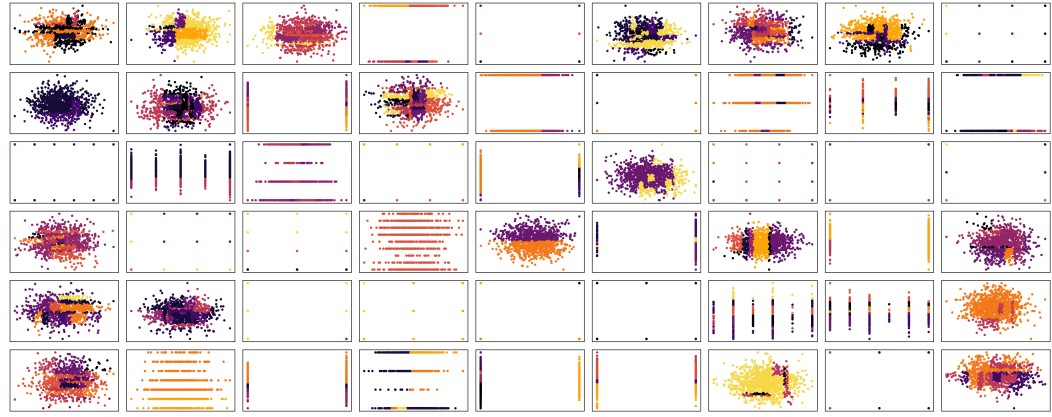

Figure 13: Generated forest data. Every box is a generated dataset with its own classes (color) and features (axes). Generated with base size 32, dataset size 1024, tree depth between 1 and 25, two features, and between 2 and 10 number of classes. See also Figures 3 and 14.

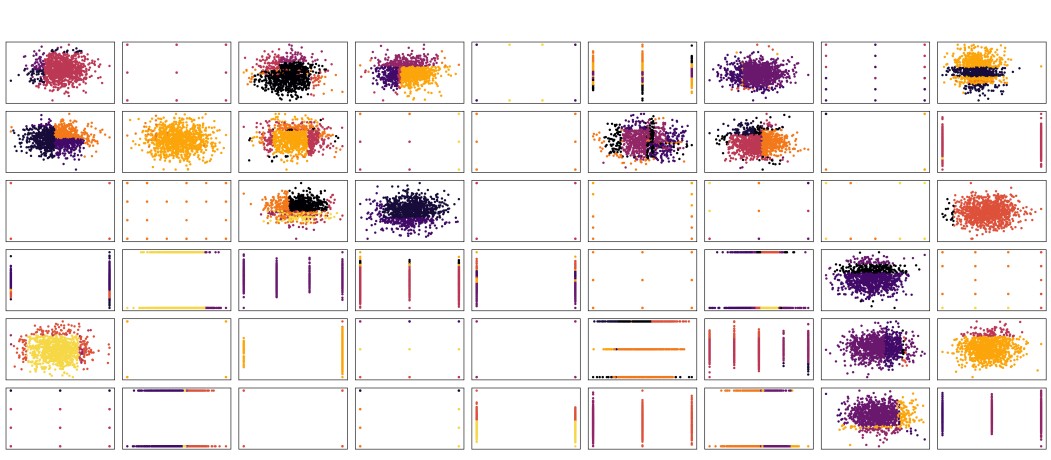

Figure 14: Generated forest data. Every box is a generated dataset with its own classes (color) and features (axes). Generated with base size 1024, dataset size 1024, tree depth between 1 and 9, two features, and between 2 and 10 number of classes. See also Figures 3 and 13.

