# OpenReview forum: "Fine-tuned In-Context Learning Transformers are Excellent Tabular Data Classifiers."
_ICLR.cc/2025/Conference — ICLR 2025 Conference Withdrawn Submission_

### Official Review · Reviewer_vRpQ · 2024-10-28

**Soundness:** 3
**Presentation:** 2
**Contribution:** 2
**Rating:** 3
**Confidence:** 3

**Summary:**

This paper extends a previous tabular data classification method TaabPFN to the finetuning setting, which tends to have a complex decision boundary. The authors have made a dataset by synthesizing data via a decision tree, which helps to show the fact that tabForest achieves better finetuning performance with more complex generated datasets. Authors mainly evaluate their method on WhyTrees and TabZilla and achieves good performance.

**Strengths:**

1. Experiments in this paper show that they achieve Sota performance on some of the datasets.
2. Authors give a visualization on the decision boundaries for the finetuned model and the zero-shot model, which helps to explain the effectiveness of their method.

**Weaknesses:**

1. Lack of novelty. This paper extends a previous model from ICL to finetuning and obtains a better performance, which is a very naive result since training (finetuning) a model always leads to better performance than using prompts (ICL), this makes the findings in the paper not meaningful.
2. Similarly, the model after training can have a more complex decision boundary than the zero-shot model, this is also a very straightforward conclusion that does not bring anything new for me.
3. Authors compare their methods with some very traditional models such as SVM, MLP, Decision Tree, XGBoost, and so on. Most of these methods have been proposed for more than ten years. A comparison of these methods can not prove that the proposed methods are good. Please compare  with more SOTA methods, and also report some other information such as the parameters, inference time for each model.
4.  The writing should be improved, especially for the figures in the paper.
5. Authors mentioned that the proposed finetuned method is an approach to reduce the requirements for GPU memory by two times, but no quantitative numbers for GPU memory required are provided in this paper.

**Questions:**

Please refer to the weakness. My major concern is the contribution (novelty) of this paper and the evaluation.

---

### Official Review · Reviewer_z5wU · 2024-11-04

**Soundness:** 2
**Presentation:** 3
**Contribution:** 3
**Rating:** 5
**Confidence:** 3

**Summary:**

This paper presents a TabPFN method for tabular data classification. They show that their TabPFN achieves better performance than traditional machine learning models, such as random forests. The authors propose TabFPN based on the complexity of the decision boundary and its relevance to the classification of tabular data.
Based on their insights on the decision boundary, the authors propose a fine-tuning method for TabPFN models to improve performance by complicating the decision boundaries. To complicate the learning of decision boundaries, they generate unrealistic data that is different from existing tabular data and use it as fine-tuning data.

**Strengths:**

1. It is interesting to see that they constructed their own dataset to complicate the decision boundary and fine-tuned with that dataset to increase performance.
2. It seems that the transformer-based model can show higher performance than the machine learning-based model. It is interesting to see a fine-tuning approach that can be used in the future to train deep neural networks on tabular data.

**Weaknesses:**

1. Although the authors' TabPFN method achieves better results than the classical tree-based machine learning models, the performance improvement is marginal. For example, in Table 3, TabPFN outperforms XGBoost by only about 0.03 to 0.04.
2. Figure1 shows that the fine-tuned model has a more complex decision boundary than the untuned model. However, when comparing the fine-tuned model and random forest, it is difficult to say that the decision boundary looks more complicated. The fine-tuned model seems to have a more complex decision boundary than the non-fine-tuned model, but when comparing it to a random forest, it is hard to say that the decision boundary is complex.

**Questions:**

1. Is there any result of training TabPFN, including the data used when fine-tuning TabForestPFN? Is it true that this data can only be used in the fine-tuning process?
2. I understand that the TabForestPFN method improves performance by affecting the decision boundary, but then I do not know if the method affects anything other than the decision boundary. It's not convincing that it only complicated the decision boundary. Authors need to explain how one can be sure that the performance improvement is due to the decision boundary.

---

### Official Review · Reviewer_dstQ · 2024-11-05

**Soundness:** 2
**Presentation:** 2
**Contribution:** 2
**Rating:** 5
**Confidence:** 4

**Summary:**

This paper studies in-context learning (ICL) for tabular data, building upon TabPFN. The authors showed that fine-tuning leads to better downstream performance, and better pre-training with more complex data (forest-based) helps the fine-tuning performance. The authors proposed a novel way to create less realistic but more complex pre-training data. The resulting pre-trained model (including both original and new pre-train data), TabForestPFN, when fine-tuned, achieves the best performance across a wide range of tabular datasets and tasks.

**Strengths:**

S1. The paper identifies that fine-tuning is still needed to achieve the best performance on each downstream task.

S2. It proposes a novel, seemingly simple-to-implement way to create more complex pre-training data. The idea to incorporate decision trees in data generation is neat, as it encourages the neural net model to mimic Tree-based models' predictions.

S3. Expensive experiments are conducted.

**Weaknesses:**

W1. The motivations of this paper are not very clear and convincing. First, it is intuitive that fine-tuning could create more complex boundaries and improve the downstream performance --- the fine-tuning process needs to fit the downstream data. Second, it is intuitive that when the downstream data are sufficient (large context), then fine-tuning would improve more. (Otherwise, it may overfit.) Third, I'm not entirely convinced that pre-training from scratch cannot generate complex boundaries. Basically, if one over-fits the model to the training data, it should create a complex enough boundary. Fourth, I didn't fully capture "We find that fine-tuned ICL-transformers, in contrast, are able to create these complex decision boundaries similar to tree-based methods." Again, it is intuitive that fine-tuning would create more complex boundaries; do ICL-transformers play an important role in this statement?

W2. It is a bit hard to digest what each step in Algorithm 1 does. Can the authors provide a visualization, for example, a series of sub-figures where each of them corresponds to the outcome of each step in Algorithm 1?

W3. Experimental results on WhyTrees show good signs of TabForestPFN (or just TabForest) but Table 3 (on TabZilla) seems to suggest no benefit of TabForestPFN over TabPFN if both are fine-tuned. Can the authors provide more details about these?

W4. I'm not sure if I completely buy the argument that TabPFN achieves better ZSL performance because its training data are more realistic. If the downstream tasks (realistic data) do require more complex boundaries, isn't it possible to create realistic but complex data?

W5. The analyses lack sufficient insights. For example, the way section 5.6 is written seems to be too intuitive: fine-tuning excels when there are sufficient samples; a larger context (i.e., more training data for the downstream tasks) improves the performance. I am curious why at larger support sizes (Figure 7 (b)), fine-tuning shows more benefits than zero-shot. It would be great if the authors could provide more insights rather than superficially summarizing the observations.

**Questions:**

I do not have further questions besides the weaknesses listed above. It would be great if the authors could address the weaknesses during the rebuttal. I'm open to adjusting my ratings based on the authors' responses.

---

### Note · Authors · 2024-11-27

**Comment:**

Dear Reviewers,

Due to the relatively low scores, we decided to allocate our time elsewhere and not further burden the reviewers with a rebuttal. We would like to thank all the reviewers for their participation.

**Withdrawal Confirmation:**

I have read and agree with the venue's withdrawal policy on behalf of myself and my co-authors.